# Complete Metabolic Response to Combined Immune Checkpoint Inhibition after Progression of Metastatic Colorectal Cancer on Pembrolizumab: A Case Report

**DOI:** 10.3390/ijms241512056

**Published:** 2023-07-27

**Authors:** Carolin Krekeler, Klaus Wethmar, Jan-Henrik Mikesch, Andrea Kerkhoff, Kerstin Menck, Georg Lenz, Hans-Ulrich Schildhaus, Michael Wessolly, Matthias W. Hoffmann, Andreas Pascher, Inga Asmus, Eva Wardelmann, Annalen Bleckmann

**Affiliations:** 1Department for Medicine A, Hematology, Oncology, Hemostaseology and Pneumology, University Hospital Muenster, 48149 Muenster, Germanyannalen.bleckmann@ukmuenster.de (A.B.); 2West German Cancer Center, University Hospital Muenster, 48149 Muenster, Germany; 3Institute of Pathology Nordhessen, 34119 Kassel, Germany; 4Institute of Pathology, University Hospital Essen, 45147 Essen, Germany; 5West German Cancer Center, University Hospital Essen, 45147 Essen, Germany; 6Department of General and Visceral Surgery, Raphaelsklinik Muenster, 48143 Muenster, Germany; 7Department of General, Visceral and Transplant Surgery, University Hospital Muenster, 48149 Muenster, Germany; 8Department of Nuclear Medicine, University Hospital Muenster, 48149 Muenster, Germany; 9Gerhard-Domagk-Institute of Pathology, University Hospital Muenster, 48149 Muenster, Germany

**Keywords:** immune checkpoint inhibition, nivolumab, ipilimumab, microsatellite instability, metastatic colorectal cancer

## Abstract

DNA mismatch repair deficient (dMMR) and microsatellite instable (MSI) metastatic colorectal cancer (mCRC) can be successfully treated with FDA- and EMA-approved immune checkpoint inhibitors (ICI) pembrolizumab and nivolumab (as single agents targeting the anti-programmed cell death protein-1 (PD-1)) or combinations of a PD-1 inhibitor with ipilimumab, a cytotoxic T-lymphocyte-associated protein 4 (CTLA-4)-targeting antibody. The best treatment strategy beyond progression on single-agent ICI therapy remains unclear. Here, we present the case of a 63-year-old male with Lynch-syndrome-associated, microsatellite instability-high (MSI-H) mCRC who achieved a rapid normalization of his tumor markers and a complete metabolic remission (CMR), currently lasting for ten months, on sequential ICI treatment with the combination of nivolumab and ipilimumab followed by nivolumab maintenance therapy after progression on single-agent anti-PD-1 ICI therapy. The therapy was well-tolerated, and no immune-related adverse events occurred. To the best of our knowledge, this is the first case of a sustained metabolic complete remission in an MSI-H mCRC patient initially progressing on single-agent anti-PD-1 therapy. Thus, dMMR mCRC patients might benefit from sequential immune checkpoint regimens even with long-term responses. However, further sophistication of clinical algorithms for treatment beyond progression on single-agent ICI therapy in MSI-mCRC is urgently needed.

## 1. Introduction

Mutations in the DNA repair genes *MLH1*, *MSH2*, *MSH6*, and *PMS2* may result in DNA-mismatch-repair-deficient cancer, characterized by a high mutational burden in repetitive DNA sequences, so-called microsatellites. Therefore, MMR deficiency is often associated with a high frequency of microsatellite instability, as found in various cancer entities [1].

While MMR deficiency is seen in about 10 to 15% of colorectal carcinomas, an underlying germline mutation in at least one DNA repair gene can be found in approximately 3% of all CRC patients [2,3]. Inherited MSI (hereditary non-polypous colorectal cancer; HNPCC, Lynch syndrome) is one of the most frequent tumor predisposition syndromes in CRC.

MSI-H mCRC responds poorly to cytotoxic treatment. Molecularly, genomic alterations in dMMR/MSI-H CRC lead to the creation of neoantigens and thus immunogenic epitopes on the cell surface. The microenvironment of dMMR/MSI-H CRC has been demonstrated to be enriched in tumor-infiltrating lymphocytes directed against these distinct tumor neoantigens, indicating ongoing immune surveillance in those cancers [4,5]. Nevertheless, an effective anti-tumor immune response is thwarted by an augmented expression of immune checkpoint molecules including PD-1, its ligand anti-programmed cell death protein-ligand 1 (PD-L1), and CTLA-4 and thus immunosuppressive features of the tumor immune ecosystem [6,7,8,9]. These immune escape mechanisms allow growth and progression of dMMR/MSI-H mCRC and are key barriers in anti-tumor immunity, as they inhibit T-cell anti-tumor responses [10].

Immune checkpoint inhibitors have revolutionized the treatment of solid tumors as they can overcome this blockade and strengthen anti-tumor immunity. For dMMR/MSI-H mCRC patients, ICI therapy has been shown to induce an effective anti-tumor immune response with long-term efficacy, while MSI stable tumors poorly respond to immunotherapy. Therefore, testing for MMR protein expression or MSI, preferably both, is mandatory.

More in detail, treatment of dMMR/MSI-H mCRC with a single anti-PD-1 agent, such as pembrolizumab (evaluated in KEYNOTE-164 [11,12] and KEYNOTE-177 trials [13]) or nivolumab (evaluated in the CHECKMATE-142 trial [14]), resulted in response rates of up to 32% and a median progression-free survival (PFS) and overall survival (OS) of 4.1/47.0 months for pembrolizumab and a 12-month overall survival rate of 73% for nivolumab, respectively [15,16]. Combined ICI therapy using the anti-PD-1 antibody nivolumab and the anti-CTLA-4 antibody ipilimumab has been demonstrated to increase the tumor infiltration with T-cells in melanomas [10]. In dMMR/MSI-H mCRC patients, the combined immune checkpoint blockade has shown increased objective response and disease control rates of 69% and 84%, respectively (median PFS/OS not reached at a minimum 24.2 months-follow-up; CHECKMATE-142 trial [17,18]). Hence, combined ICI treatment is already the current standard of care for MSI-H/dMMR mCRC patients in a first-line setting in the US and approved by the FDA.

However, the majority of patients with an initial response to single-agent ICI eventually develop progressive disease due to acquired ICI resistance [9,10,19,20,21,22]. The mechanisms of secondary resistance to ICI treatment are diverse and entity-specific [9]. Infiltration with tumor-suppressive cells, downregulation of major histocompatibility complexes (MHC) class I molecules, and thus, hampered antigen presentation, apotosis inhibition by hypoxia, alterations in gut microbioma, defective tp53 and INF-signal pathways, amplified WNT/ß-Catenin, and TGF-ß signaling, inducing an increased portion of cancer stem cells (CSCs) that impact the tumor environment, are factors that promote to the tumor-mediated ICI bypass [9,10,20].

With rising evidence to the underlying mechanisms of hampered ICI efficacy, therapeutic approaches to restore the immune system and boost ICI efficacy, such as CSC-directed therapy or fecal microbial transplantation, are increasing, but currently remain experimental.

At present, there is no general recommendation for further treatment options, and especially not for an adjustment of the immunotherapeutic regimen in dMRR/MSI-H mCRC patients progressing on single-agent anti-PD-1 ICI therapy.

Here, we report on the clinical course of a patient with KRAS-mutated dMMR/MSI-H mCRC successfully treated with dual checkpoint inhibition using nivolumab and ipilimumab after initial progression under single anti-PD-1 agent pembrolizumab.

## 2. Case Presentation

A 63-year-old male was diagnosed with a pT3, pN1c, pMx, L0, V0, Pn0, UICC (version 8) stage IIIB high-grade (G3), KRAS-mutated (Q61K) adenocarcinoma of the ascending colon in December 2019. Figure 1 illustrates the relevant timepoints of the diagnostic assessments and treatment strategies.

Histopathological work-up revealed the loss of MSH6 expression in the tumor tissue (Figure 2), and germline testing detected a frame-shift mutation in *MSH6 gene*, implicating Lynch syndrome. The patient was treated with four adjuvant courses of XELOX (capecitabine and oxaliplatin) after initial right hemicolectomy prior to initiating further care in our department. Post-treatment imaging showed no further evidence of disease.

Eight months prior to the diagnosis of the dMMR/MSI-H colon carcinoma, the patient had already been diagnosed with prostate carcinoma (initial Gleason-Score 3 + 3 = 6) and high-grade prostate intraepithelial neoplasia (PIN). The lesions were regularly monitored via MRI and 3-monthly biopsies were performed while chemotherapy with capecitabine and oxaliplatin (XELOX) was applied. In July 2020, biopsies revealed a rising Gleason score (3 + 5 = 8), and the patient underwent robotic-assisted radical prostatectomy. As expected from the germline variants observed in the colon cancer samples, the immunohistochemical workup also revealed a loss of MSH6 expression in the prostate cancer tissue. Postoperative PSA-levels were always normal.

In November 2020, the first follow-up imaging by computed tomography (CT) showed multiple liver nodules, suspicious for diffuse liver metastases. At that time, the patient was admitted to our emergency department with melaena, infrapubic swelling, and severe periumbilical pain. Colonoscopy and balloon-assisted enteroscopy were inconclusive, while positron emission tomography and computed tomography (PET/CT, Figure 3a) showed local tumor recurrence at the site of the ileocolic anastomosis as well as massive disease progression with diffuse peritoneal carcinomatosis, subcutaneous lesions in the lower left abdomen at former laparoscopic port sites, one bone lesion in the tenth rib on the right, and multiple metastases to the liver and abdominal lymph nodes.

Treatment with pembrolizumab was initiated at a fixed dose of 200 mg every three weeks (q3). Pembrolizumab was well tolerated, brought a rapid clinical improvement after the first dose with a decrease in abdominal swelling and pain, and reduced levels of the tumor markers carcinoembryonic antigen (CEA) and carbohydrate antigen 19-9 (CA 19-9, see Figure 4). A CT scan after three weeks of treatment confirmed the partial remission of the metastatic lesions (images not shown). However, after seven cycles of pembrolizumab monotherapy the CEA and CA 19-9 levels increased again to 19 ng/mL and 687 U/mL, respectively, and the patient developed indurated painful subcutaneous metastases of the abdominal wall. PET/CT imaging confirmed the suspected disease progression (Figure 3b).

In May 2021, the patient began a four-course treatment with the combination of the anti-PD-1 antibody nivolumab (3 mg/kg body weight) and the anti-CTLA-4 antibody ipilimumab (1 mg/kg body weight). This was chosen in view of the rapid initial clinical and serological response to pembrolizumab and the long-lasting clinical benefit seen with the combined blockade of PD-1 and CTLA-4 [14], which had just been approved for treatment of MSI-high CRC after failure of one line of chemotherapy [17]. The treatment was well tolerated, except for initial severe periumbilical pain that was controlled by oral opioids, and an inflammation of the abdominal wall after the third cycle of treatment that was thought to be immune-mediated. With this suspicion in mind, 10 mg oral prednisolone per day was applied together with the fourth cycle of combined ICI therapy. CA 19-9 and CEA levels declined rapidly, the latter returning to normal after three therapy cycles. A PET-CT scan revealed partial remission with a complete metabolic remission of the bone lesion and regression of all other metastases (Figure 3c).

After four cycles of nivolumab and ipilimumab, the patient was admitted to the emergency room with clinical signs of an acute abdomen showing elevated C-reactive protein (CRP) levels (20.5 mg/dL; normal range < 0.5 mg/dL) and an indurated abdominal swelling caused by abscesses in the abdominal wall at the sites of the former subcutaneous and peritoneal metastases. The abscesses were drained, intravenous antibiotic treatment was initiated, and the condition of the patient improved. The nivolumab maintenance therapy was continued at a constant dose of 240 mg every three weeks (q3w). However, a few days later, the patient was re-admitted to the hospital with progressive swelling of the right abdominal wall, rising CRP levels, and severe abdominal pain. An MRI with an oral contrast agent revealed a phlegmonous abdominal wall with multiple enteral fistulas and abscesses at sites of former metastases. That condition was most likely caused by rapid, treatment-induced tumor necrosis with subsequent superinfection. The abscesses were drained, dead and infected tissue was surgically debrided, and a combined antimycotic and antibiotic treatment was administered. Under the combined interventional and conservative treatment, the patient’s condition improved, and the laboratory parameters returned to normal.

Although desired by the patient, the surgical excision of the fistulas and reconstruction of the abdominal wall was not recommended due to the marked local inflammation and the ongoing immunotherapy with nivolumab.

With the patient’s consent and on his insistence, a laparotomy was performed, during which the perforated small bowel sections and the fistulas were removed and the abdominal wall was reconstructed (Figure 5).

The histopathological work-up showed necrosis in former tumor lesions and one vital adenocarcinoma of the small intestine, again with an immunohistochemical-confirmed loss of MSH6 expression. The postoperative PET-CT scan (Figure 3d) indicated a persisting therapeutic response, although with a new metabolically active lymph node in the surgical site (SUV_max_ 34.6) that was interpreted as a postoperative reactive lymphadenopathy.

After nine cycles of single-agent nivolumab 240 mg, q2w the condition of the patient was excellent, the tumor markers remained normal, and a PET-CT confirmed an ongoing PR in March 2022 (Figure 3e). The nivolumab maintenance therapy was, therefore, adjusted to a four-week interval with a flat dose of 480 mg. In September 2022, during an incisional hernia repair, no signs of a relapse were observed. In addition, after a total duration of 17 months of second-line dual ICI therapy, including 14 months of nivolumab maintenance therapy, PET-CT imaging (Figure 3f) first showed a complete metabolic remission (CMR) and no morphological evidence of residual disease in September 2022. Since then, the patient has remained in ongoing complete metabolic remission with tumor markers CEA and CA 19-9 within normal ranges during the 24-month follow-up after initiation of relapse-directed treatment.

To further investigate the underlying causes and mechanisms of this excellent response to combined anti-CTLA-4 and anti-PD-L1 blockade after progression under single agent PD-1 inhibiting therapy, we performed a molecular pathological analysis of the tumor tissue. We found no distinct *CD274*/PD-L1 mutation, which might possibly have reduced the efficacy of pembrolizumab, but not that of nivolumab due to the different binding sites. The tumor harbored a JAK2-mutation (R1063H, ClinVar: CN169374), which is described as a gain of function mutation (https://ckb.jax.org/geneVariant/show?geneVariantId=7650 (accessed on 29 June 2023)). In addition, 12 HLA class I somatic mutations were found in the tumor tissue (Table 1). Individually, none of the mutations have been described as pathogenic variant in molecular analysis databases. However, the frequency of mutations at the *HLA* gene loci was increased by a factor of 7.5 compared with the expected frequency of random gene mutations (*p* < 0.01). Nonetheless, the impact of the combination of several *HLA* mutations, and especially an eventual upregulation of *HLA* class I expression on antigen-presenting cells by the mutations, remains elusive.

## 3. Discussion

Here, we present a case of a complete metabolic and serologic remission in a patient with dMMR/MSI-H mCRC and Lynch syndrome-associated prostate cancer under treatment with combined anti-PD-1 and anti-CTLA-4 blockade with nivolumab and ipilimumab after progression under pembrolizumab monotherapy.

Treatment of mCRC with dMMR/MSI-H has traditionally been challenging due to the poor response to conventional chemotherapeutic therapy [23,24]. The development of FDA- and EMA-approved single agent immunotherapeutic drugs targeting PD-1, such as nivolumab and pembrolizumab, has revolutionized the therapeutic strategies in dMMR/MSI-H mCRC, refractory or relapsed after cytotoxic treatment, due to high objective response rates (ORR) and sustained responses [11,14]. Compared to the therapy with a single agent anti-PD-1 antibody, the combination of nivolumab (targeting PD-1) and ipilimumab (targeting CTLA-4) applied in the phase II CheckMate-142 trial yielded an increased ORR of 55% in patients previously treated with chemotherapy and showed a manageable safety profile [14,25]. Thus, dual checkpoint blockade was approved by the FDA in a first-line setting and by the EMA in dMMR/MSI-H mCRC after prior fluoropyrimidine-based treatment.

The use of pembrolizumab as a first line treatment in dMMR/MSI-H mCRC was shown to be superior to 5-fluoruracil-based therapy with regard to progression-free survival [19]. However, the combination of nivolumab and ipilimumab studied in the CHECKMATE-142 trial [17] showed a superior and durable responses, and thus, constitutes the new first-line standard treatment in dMMR/MSI-H mCRC in the US, but is not yet authorized by the EMA.

Treating patients with progression after initial response to immune checkpoint inhibitors remains a major challenge. As every antibody directed against PD-1 binds to a distinct epitope, sequential therapy with another PD-1 blocker after progression on an anti-PD-1 antibody may be an option, although clear evidence for antibody-specific mechanisms of resistance is lacking [26,27]. Recent data for such sequential treatment with a single-agent anti-PD-1 drug or a combined use with a CTLA-4 inhibitor showed non-uniform responses ranging from durable response to massive disease progression [28,29,30,31].

Analysis of tumors with acquired resistance to anti-PD-1 antibodies demonstrated epigenetic changes, such as histone marks, DNA methylation, and miRNA signatures, eventually leading to an altered microenvironment and immune escape [32]. Furthermore, there is evidence that the tumor cells can escape immune surveillance by upregulating immune-suppressive molecules including lymphocyte-activation gene (LAG-3) and CTLA-4 as well as by the absence or decreased infiltration of (CD-8 positive) tumor-infiltrating leukocytes or the increased presence of immunosuppressive cells, such as regulatory T cells and myeloid-derived suppressor cells, in the tumor microenvironment [33,34]. Thus, the addition of a second immune checkpoint inhibitor like the anti-CTLA-4 antibody ipilimumab may help to overcome acquired resistance to the PD-1 blockade. There are emerging data showing long-term disease control after re-challenging immunotherapy with CTLA-4 and PD-1 antibodies in patients whose tumors had progressed on single agent PD-1 blockade in melanoma [35,36], hepatocellular carcinoma [37], non-small cell lung cancer [38], and urothelial cancer [39].

Moreover, loss of antigen expression, for example via defects in interferon-γ pathway due to mutations in JAK2-signaling [40], is an alternative mechanism of immune escape. We found one pathogenic somatic gain-of-function mutation in our patient. The mutation variant (*JAK2 R1063H*) lies within the protein kinase domain 2 of the JAK2-protein and results in, among other things, increased transcriptional activity. Gain of function mutations in JAK2-signaling are described as being associated with a significant increase in PD-L1 expression, which can affect response to immunotherapy [41,42].

Recently, downregulation of HLA class-I molecules was described as being associated with an inferior response to ICI therapy due to the impaired interaction with the antigen-presenting machinery at the protein level [43,44,45]. On the other hand, high HLA evolutionary divergence, and thus, the physiochemical sequence divergence between HLA-I alleles, has not only been shown to enable the presentation of more diverse immunopeptidomes, but to be associated with better survival [46,47]. The distinct HLA phenotype resulting from the various HLA mutations observed in our patient might have, thus, possibly led to (a) HLA protein defects resulting in alternative/increased epitope presentation, (b) an upregulation of the HLA-expression itself, or (c) a higher evolutionary divergence, and thus, increased diversity of neoepitopes. Hence, the patients’ HLA variant might have conceivably contributed to higher immunogenicity and thus to an improved response to combined checkpoint inhibition compared with anti-PD-1 monotherapy. Moreover, it may have been crucial for the sustained CMR of this case of metastasized HNPCC-associated MSH6 deficient CRC under dual immune blockade. Future work on larger patient cohorts is required to resolve the predominant molecular mechanisms predicting response to dual checkpoint-inhibition after progress under PD-1 blockade.

In the literature, we found two other cases of dMMR/MSI-H CRC treated with sequential combined nivolumab and ipilimumab therapy after progression on single-agent pembrolizumab. In line with our findings, the re-challenge with immunotherapy led to partial responses of patients with dMMR/MSI-H mCRC in other case studies [48,49]. Winer et al. treated a 64-year-old patient with Lynch syndrome associated mCRC and urothelial carcinoma with the anti-PD-L1 antibody atezolizumab after progression on pembrolizumab and subsequently with ipilimumab and nivolumab after clinical progression marked by rising tumor markers. Even if the patient had similar clinical features (age, Lynch-syndrome, and multiple HNPCC-associated cancer entities) and a lower tumor burden (liver metastases only) compared to our patient, only a partial metabolic response and stabilized but strongly elevated CEA levels were reached [49]. Das et al. reported a 30% tumor mass reduction in an MSI-H mCRC female patient (Lynch-syndrome association not indicated) after four cycles of combined checkpoint inhibition and nivolumab maintenance therapy [48].

To our knowledge, this is the first published case describing a PET-CT-confirmed metabolic complete remission and serologic normalization of tumor markers CEA and CA 19-9. The causes for this excellent response, especially in comparison to the patient case reported by Winer et al., with a patient comparable in disease manifestation and demographics, remain uncertain. Possibly, the patient’s good performance status at most points of the therapy, limited pre-existing illnesses (ischemic stroke without remains, hyperlipidemia), and the limited co-medication (ASS, rosuvastation, ezetimib) have contributed to an undelayed therapy administration and might have, thus, also contributed to this excellent therapeutic response. Moreover, the ethnicity of our patient might have influenced the response to ICI therapy, as non-Hispanic white patients are known to show higher response rates [50,51]. Hence, demographics and treatment characteristics might have impacted the therapy response. Finally, following a single case description, the results may not automatically be applied to a larger cohort.

Molecular pathologic analyses and examination for MSI are implemented in the clinical and diagnostic algorithms of mCRC. During the last decade, a variety of characteristic mutations resulting in signal pathway activation, and thus, driving tumor growth, became targetable. This case study underlines the high importance of molecular analyses and a patient-individual treatment regimen based on the underlying tumor biology. Precision medicine, and thus, the combination of drugs based on molecular tumor characteristics, will become increasingly important in the foreseeable future. As they can provide tumor control even in advanced therapy lines, clinicians should assess the molecular characteristics in mCRC for potential molecularly based therapy options as for the patient in our case study.

ICI treatment has superseded conventional cytotoxic treatment in MSI-H mCRC and has been implemented as a first-line standard in current guidelines. Despite the limitations of a single case report mentioned here, the rapid and ongoing therapy response in overcoming the acquired PD-1 resistance by adding an anti-CTLA-4 antibody may be a promising strategy for other patients with Lynch syndrome-associated CRC. However, for the inclusion of sequential ICI treatment beyond progression on single-agent anti-PD-1 therapy, future clinical trial evaluation is warranted, as patients with disease progression on single-agent anti-PD-1 therapies have been excluded in recent trials and especially in the CHECKMATE-142 trial.

The combination of drugs seems to reduce the immune escape phenomena associated with a single-target directed ICI therapy as multiple resistance mechanisms are required, achieves higher ORR and will, thus, become the first-line standard in the first-line setting in the foreseeable future [18].

## 4. Conclusions

Post-progression treatment of dMMR/MSI-H mCRC on anti-PD-1 therapy remains a major challenge. The addition of an anti-CTLA-4 drug might help to overcome initial or acquired resistance to ICI monotherapy and is able to provide a durable response this patient collective. Our patient with Lynch syndrome treated with four courses of nivolumab/ipilimumab combined and subsequent nivolumab maintenance therapy achieved an ongoing complete metabolic remission with normalized tumor markers.

Even if the predominant mechanisms and molecular features leading to the high effectiveness of sequential double-immune checkpoint blockade in individual patients are not yet fully understood, specific HLA mutations have emerged as biomarkers for the ICI response. The high frequency of HLA mutations observed in the tumor sample of our patient might have contributed to the excellent therapeutic response. Prospective HLA-genotyping may help to better understand the biological impact of distinct single mutations, and especially the combined effects of multiple HLA aberrations.

Prospective clinical trials are needed to confirm the effectiveness and safety of combined immunotherapy after progression on single agent ICI in dMMR/MSI-H mCRC, comparing sequential ICI therapy to a re-initiation of conventional chemotherapy, and to evaluate the significance of HLA gene mutations in predicting responses to checkpoint blockade.

## Figures and Tables

**Figure 1 ijms-24-12056-f001:**
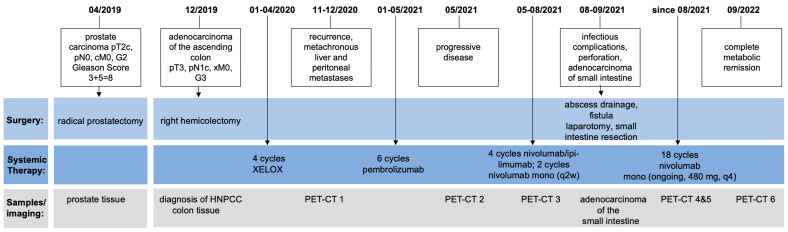
Timeline of the patient’s treatment and course of the disease including obtained samples and imaging performed.

**Figure 2 ijms-24-12056-f002:**
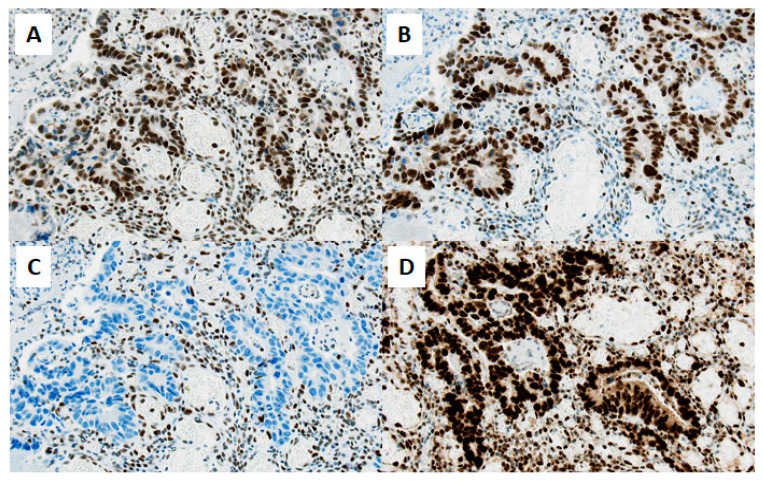
Immunohistochemistry reveals loss of MSH-6 mismatch repair protein in the tumor tissue immunohistochemical staining of mismatch repair proteins in tumor tissue of the small intestine shows an isolated nuclear loss of MSH6 (**C**), whereas MSH2 (**A**), MLH1 (**B**), and PMS2 (**D**) are strongly expressed (original magnification ×200, scale bar ≅ 50 µm). Nuclear staining was assessed in comparison to lymphocytes serving as internal positive control.

**Figure 3 ijms-24-12056-f003:**
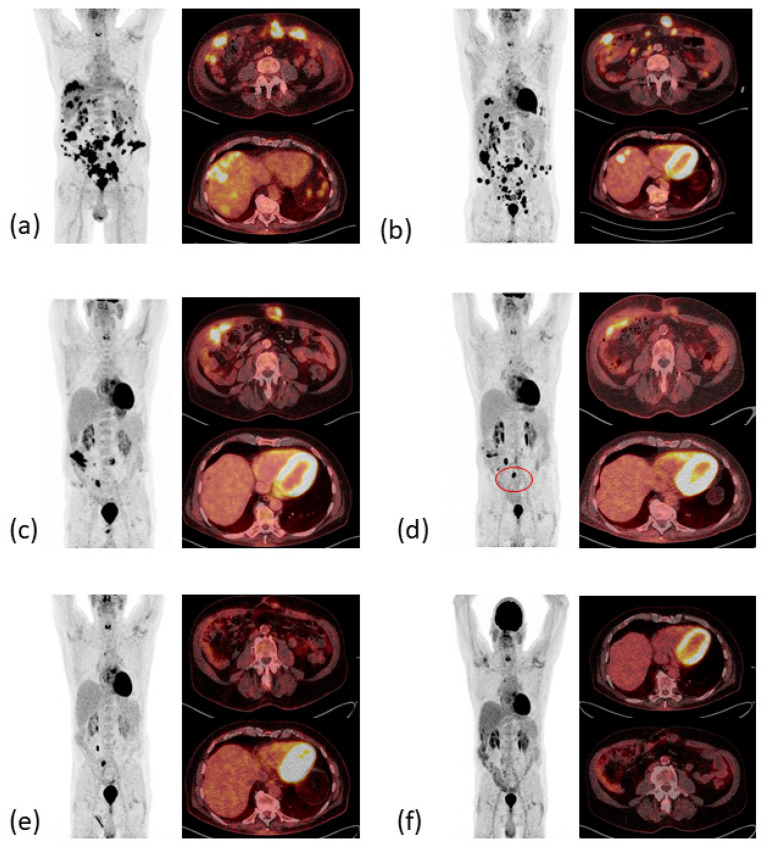
PET-CT images therapeutic response resulting in complete metabolic remission. (**a**) Local relapse and metachronous metastatic disease (December 2020). (**b**) Mixed response in April 2021 after six cycles pembrolizumab. (**c**) PR with decrease of metabolic activity and complete metabolic remission (CMR) of liver and bone metastases in June 2021. (**d**) Sustained CMR of liver metastases. Postoperative increased metabolic activity of one lymph node (red circle) in November 2021. (**e**) PR of the subcutaneous metastases, sustained CMR of liver metastases in March 2022. (**f**) CMR in September 2022.

**Figure 4 ijms-24-12056-f004:**
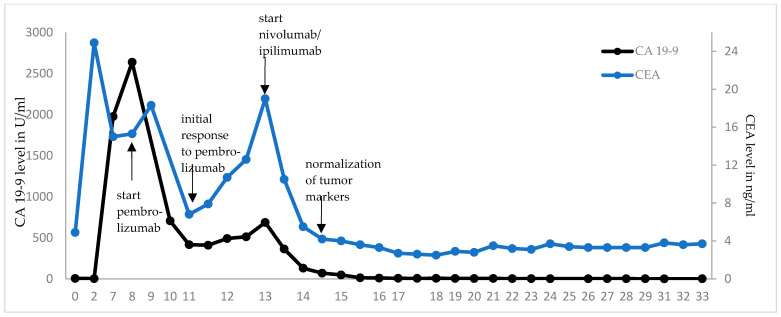
Dynamics of tumor markers after initiation of ICI therapy. CEA and CA 19-9 were monitored in regular blood draws. Rising tumor marker levels were seen after initial decrease under pembrolizumab monotherapy, matching with the image-confirmed disease progression. Treatment with ipilimumab and pembrolizumab led to a rapid decline and eventually normalization of CEA and CA 19-9 levels, respectively. Under ongoing nivolumab monotherapy, both markers are persistently in normal ranges.

**Figure 5 ijms-24-12056-f005:**
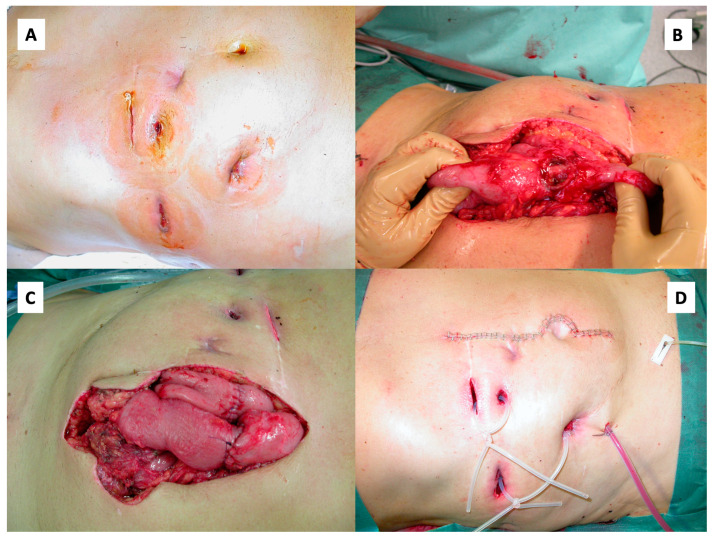
Laparotomy, fistula excision and abdominal wall reconstruction. Abdomen prior to surgery with multiple entero-cutaneous fistulas. (**A**) Necrotic former tumor tissue in the small intestine (**B**). After fistula excision, multiple end-to-end enterostomies were performed (**C**). Finally, the abdominal wall was reconstructed (**D**).

**Table 1 ijms-24-12056-t001:** Multiple *HLA* mutations were observed during molecular pathological analysis of colon tumor tissue, which might contribute to the high efficacy of combined ICI therapy in this patient. All variants are, individually, of benign or indeterminate potential.

Locus (Transcript ID)	Start Position	End Position	Type of Variant	Affected DNA Section	Encoded Gene
Chr.6 p22.1 (ENST00000396634.5)	Substitution				
	29726903	29726904	AA > CT	ncRNA	
	29943483	29943484	AC > CG	Exon	*HLA-A*
	29943494	29943495	GT > CG	Exon	*HLA-A*
	29944153	29944154	CA > TG	Exon	*HLA-A*
	SNV				
	29944118	29944118	T > A	Intron	*HLA-A*
	29944151	29944151	C > G	Exon	*HLA-A*
Chr. 6 p21.33 (ENST00000376228.9)	Substitution				
	31271152	31271154	CAG > GTC	Exon	*HLA-C*
	31356226	31356227	TC > GT	Promotor	
	31356246	31356248	CCG > GTC	Promotor	
	31356748	31356749	CC > TG	Promotor	
	SNV				
	31271082	31271082	G > C	Exon	*HLA-C*
	31271089	31271089	C > G	Exon	*HLA-C*

## Data Availability

The data presented in this study are available on request from the corresponding author. The data are not publicly available due to privacy reasons.

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
