# Peer review of "Complete Metabolic Response to Combined Immune Checkpoint Inhibition after Progression of Metastatic Colorectal Cancer on Pembrolizumab: A Case Report"

_ijms, 2023, doi:10.3390/ijms241512056_

Round 1

Reviewer 1 Report

The manuscript entitled “Complete metabolic response to combined immune checkpoint inhibition after progression of metastatic colorectal cancer on pembrolizumab: A case report” is a case report reporting probably the first case of a sustained metabolic complete remission in an MSI-H mCRC patient initially progressing on single-agent anti-PD-1 therapy. The case report is an appreciable work. However, the manuscript needs some improvements.

1. The authors must work on formatting and typo/spacing errors.

2. The abstract is not well written. It may be concise and written in a structured form.

3. The discussion part must incorporate a comparative discussion concerning the outcome of the case report and previous studies (for example, references 18 and 19).

4. The authors need to discuss more limitations (age, sex, ethnicity, polypharmacy, and so on) of the study and mention the possible impact of these limitations on the study outcome.

5. This is a single case study. Discussing the foreseeable therapies (a combination of drugs) that can provide similar results in the given patients will be valuable.

6. Discussing the outcomes of the case study and precision medicine may be a good addition for the readers.

7. The conclusion should not contain similar information as the introduction. Given the importance of the case study findings and main points as puzzle pieces, the conclusion needs rewriting.

Minor editing of English language required

Author Response

The manuscript entitled “Complete metabolic response to combined immune checkpoint inhibition after progression of metastatic colorectal cancer on pembrolizumab: A case report” is a case report reporting probably the first case of a sustained metabolic complete remission in an MSI-H mCRC patient initially progressing on single-agent anti-PD-1 therapy. The case report is an appreciable work. However, the manuscript needs some improvements.

Authors: We kindly thank the author for his/her kind remarks and the positive feedback to our paper.

  1. The authors must work on formatting and typo/spacing errors.

Authors: We would like to thank the reviewer for this important point and improved the formatting in our manuscript and double-checked for typing and spacing errors. 

  1. The abstract is not well written. It may be concise and written in a structured form.

Authors: We thank the reviewer for outlining this important aspect. We re-wrote the abstract in a more structured form and highlighted the findings of our case study.

  1. The discussion part must incorporate a comparative discussion concerning the outcome of the case report and previous studies (for example, references 18 and 19).

Authors: We thank the reviewer for this important point. We added a comparison of the outcome of the patient in our case study to that in previously published case studies in lines 411-428. We highlighted the comparable patient characteristics especially in the case published by Winer et al. and discussed the potential causes for the excellent response of our patient.

  1. The authors need to discuss more limitations (age, sex, ethnicity, polypharmacy, and so on) of the study and mention the possible impact of these limitations on the study outcome.

Authors: We share the reviewer`s opinion on a hampered generalization of our study data to a larger patient cohort. We discussed on more limitations in lines 429-437 and commented in the discussion as following “Finally, following a single case description, the results may not automatically be applied to a larger cohort.”

  1. This is a single case study. Discussing the foreseeable therapies (a combination of drugs) that can provide similar results in the given patients will be valuable.

Authors: Again, we are thankful for this important annotation. We share the reviewer`s opinion on the limited generalizability of the therapy result of our patient. However, the results of the CHECKMATE-142 trial (Lenz et al., 2022) underlined the high efficacy of a combination of ICI drugs, which will establish the future first-line standard in the treatment of MSI-H mCRC. The conventional cytotoxic therapy will thus further lose importance in this collective. We commented on this in lines 448-449 and 457-460.

  1. Discussing the outcomes of the case study and precision medicine may be a good addition for the readers.

Authors: Again, we thank the reviewer for outlining this important aspect. We commented on the necessity of an individual molecular-based therapy as a precision medicine in lines 438-447.

  1. The conclusion should not contain similar information as the introduction. Given the importance of the case study findings and main points as puzzle pieces, the conclusion needs rewriting.

Authors: We would like to thank the reviewer for this important point. We revised the conclusion   focusing on the excellent patient outcome and the potentially underlying molecular mechanisms.

Authors: We very appreciate the positive feedback from the reviewer and would like to thank him/her for diligent work on our manuscript.

Reviewer 2 Report

The study by Krekeler and colleagues is a good case report demonstrating novel findings and efficient approaches conducted on a patient. The approaches and findings of the study are satisfying and I can suggest it for publication. However, the quality of the presentation is weak and requires vigorous editing. 

1- The abstract is too long and requires focusing on the achievements.

2- All abbreviations require explanation the first time they come within the manuscript

3- The introduction lacks a dynamic flow. The authors are recommended to explain the mechanisms and successes as well as challenges of cancer immunotherapies (please see PMID: 36577330, PMID: 36309224, and PMID: 36306594)

Requires improvement

Author Response

The study by Krekeler and colleagues is a good case report demonstrating novel findings and efficient approaches conducted on a patient. The approaches and findings of the study are satisfying and I can suggest it for publication. However, the quality of the presentation is weak and requires vigorous editing. 

Authors: We kindly thank the author for his/her kind remarks and the positive feedback to our paper.

  • The abstract is too long and requires focusing on the achievements.

Authors: We thank the reviewer for outlining this important aspect. We re-wrote and shortened the abstract in a more structured form and highlighted the achievement of our case study.

  • All abbreviations require explanation the first time they come within the manuscript

Authors: We double-checked all abbreviations and adapted the missing explanations at the first mentioning in the manuscript. We thank the reviewer for his/her kind advice.

  • The introduction lacks a dynamic flow. The authors are recommended to explain the mechanisms and successes as well as challenges of cancer immunotherapies (please see PMID: 36577330, PMID: 36309224, and PMID: 36306594)

Authors: We are very thankful to the reviewer for her/his important improvement suggestion. We revised the introduction and included mechanisms and successes of immunotherapy in colorectal cancer as well as challenges and resistance mechanisms. To this purpose, we also incorporated the remarkable literature proposed by the reviewer.

Authors: Once again, thank you very much to the reviewer for the suggested improvements.

Reviewer 3 Report

The authors present an interesting study describing the role of combination therapy pembrolizumab/ CHK inhibitor in the treatment of metastatic colorectal cancer. This paper could be accepted, but some minor revisions are required:

- in the introduction, the authors discuss the recent advances in immune checkpoint inhibitor combination strategies for microsatellite-stable colorectal cancer. To give more information, I suggest introducing the following recent reference, along with your reference 17 (at line 89):

iScience 2021, 24(6),102664; https://doi.org/10.1016/j.isci.2021.102664

Front Oncol. 2023, 13, 1112276; https://www.ncbi.nlm.nih.gov/pmc/articles/PMC9932591/

- FIGURE 3: the caption (A) should not be bold.

- FIGURE 5: explain the four plots (A-D) in the caption.

- A graphical abstract must be provided.

- References should be checked again (e.g. add DOI at reference [6])

Minor editing of English language required

Author Response

The authors present an interesting study describing the role of combination therapy pembrolizumab/ CHK inhibitor in the treatment of metastatic colorectal cancer. This paper could be accepted, but some minor revisions are required:

Authors: We are very thankful for this positive feedback to our paper.

- in the introduction, the authors discuss the recent advances in immune checkpoint inhibitor combination strategies for microsatellite-stable colorectal cancer. To give more information, I suggest introducing the following recent reference, along with your reference 17 (at line 89): 

iScience 2021, 24(6),102664; https://doi.org/10.1016/j.isci.2021.102664

Front Oncol. 2023, 13, 1112276; https://www.ncbi.nlm.nih.gov/pmc/articles/PMC9932591/

Authors: We highly appreciate the recommendation of these two decent publications, which we have added to our references.

- FIGURE 3: the caption (A) should not be bold.

Authors: We thank the reviewer for his/her comment. We double checked Figure 3 but could not find a bold caption (A). We hope, the Figure is now suitable for publication.

- FIGURE 5: explain the four plots (A-D) in the caption.

Authors: We are very thankful for this remark. Due to the unfavorable positioning of the caption in the initial version, the caption was separated from the figure. We now adjusted the formatting to clarify this.

- A graphical abstract must be provided.

Authors: We highly appreciate this meaningful suggestion and applied a graphical abstract. With this, we hope to provide a more appealing presentation of our case study.  

- References should be checked again (e.g. add DOI at reference [6])

Authors: We adjusted the references in our manuscript and hope that the formatting is now

 suitable for the publication.

Authors: We again thank the reviewer for his/her kind remarks and the suggestion for an acceptance of our paper.